# Flux Calculation for Primary Metabolism Reveals Changes in Allocation of Nitrogen to Different Amino Acid Families When Photorespiratory Activity Changes

**DOI:** 10.3390/ijms25158394

**Published:** 2024-08-01

**Authors:** Nils Friedrichs, Danial Shokouhi, Arnd G. Heyer

**Affiliations:** Institute of Biomaterials and Biomolecular Systems, University of Stuttgart, Pfaffenwaldring 57, 70569 Stuttgart, Germanydanial.shokouhi@bio.uni-stuttgart.de (D.S.)

**Keywords:** photorespiration, photosynthesis–respiration interactions, plant carbon and nitrogen metabolism, alternative metabolic pathways, Arabidopsis, *hpr1-1*

## Abstract

Photorespiration, caused by oxygenation of the enzyme Rubisco, is considered a wasteful process, because it reduces photosynthetic carbon gain, but it also supplies amino acids and is involved in amelioration of stress. Here, we show that a sudden increase in photorespiratory activity not only reduced carbon acquisition and production of sugars and starch, but also affected diurnal dynamics of amino acids not obviously involved in the process. Flux calculations based on diurnal metabolite profiles suggest that export of proline from leaves increases, while aspartate family members accumulate. An immense increase is observed for turnover in the cyclic reaction of glutamine synthetase/glutamine-oxoglutarate aminotransferase (GS/GOGAT), probably because of increased production of ammonium in photorespiration. The *hpr1-1* mutant, defective in peroxisomal hydroxypyruvate reductase, shows substantial alterations in flux, leading to a shift from the oxoglutarate to the aspartate family of amino acids. This is coupled to a massive export of asparagine, which may serve in exchange for serine between shoot and root.

## 1. Introduction

Respiration in light and photorespiration (PR) strongly influence the net carbon gain of plants, but also have an important impact on nitrogen assimilation [1]. This relates not only to the provision of reducing equivalents in the form of NADH, required for nitrate reductase in green tissues, but also to the supply of carbon skeletons for the fixation of ammonia that stem from TCA cycle intermediates [2]. Although flux through the TCA cycle is reduced during the light phase [3], sections of the pathway operate to provide 2-oxoglutarate to supply the glutamine synthetase/glutamate synthase (GS/GOGAT) cycle [4]. A substantial proportion of the ammonium used for glutamine (Gln) synthesis is produced by PR, which is initiated by oxygenation of ribulose-1,5-bisphosphate (RuBP) through the action of the enzyme ribulose-bisphosphate carboxylase/oxygenase (Rubisco).

While the reactions of the photorespiratory pathway are essential to detoxify 2-phosphoglycolate, a product of the oxygenation reaction [5], PR has long been considered wasteful, because it consumes ATP and reducing equivalents and, thus, reduces photosynthetic performance [6]. However, high atmospheric carbon dioxide (CO_2_) concentrations that suppress PR lead to reduced protein content and a reduction in net photosynthesis when applied over long time periods. This so-called acclimation of photosynthesis to elevated CO_2_ has stimulated investigations of possible benefits of PR for primary plant metabolism. It turned out that a functional photorespiratory metabolism protects plants from photooxidative stress imposed by high light or ozone, possibly by consuming excess reducing power [7], but it also appears to serve as a low-carbon containing store for assimilated nitrogen during the light phase, when carbon skeletons are needed for de novo N- acquisition [8]. However, the decline in atmospheric CO_2_ concentration about 35 million years ago stimulated evolution of an alternative photosynthetic mechanism, the C4 photosynthesis, that confines Rubisco to bundle sheath cells, where CO_2_ levels are high and/or oxygen levels are low and thus prevents the oxygenation reaction. Similarly, in many algae that are confronted with low CO_2_ concentration and slow diffusion in water, Rubisco is aggregated into a micro-compartment with elevated CO_2_ concentration, the pyrenoid [9]. This demonstrates that PR is not essential, when oxygenation of RuBP is prevented and some metabolic adjustments are made.

In C3 plants with high PR activity, the amino acids glycine (Gly) and serine (Ser) are predominantly synthesized in the photorespiratory pathway in leaves, while in non-green tissues the so-called phosphorylated pathway of serine biosynthesis (PPSB) prevails. This pathway starts from glycolytic 3-phosphyglycerate and goes via 3-phosphohydroxypyruvate and 3-phosphoserine [10]. In C4 plants, the PPSB dominates and, as a consequence, a larger proportion of Ser synthesis seems to occur in roots of C4 plants of the genus *Flaveria* as compared to leaves [11]. This appears to strongly influence sulfur assimilation and synthesis of cysteine (Cys), which are as well preferentially localized to roots of C4 *Flaveria* species [11], while in C4 monocots, sulfur assimilation is confined to bundle sheath cells [12]. Surprisingly, genes for enzymes involved in sulfur assimilation are also expressed in bundle sheath cells of the C3 model plant *Arabidopsis thaliana*, and thus it is unclear whether more general rules could be derived from the observations made so far [13]. 

To get a more comprehensive picture of the impact of PR on the homeostasis of primary metabolism, we applied mathematical modeling of carbon and nitrogen fluxes into and out of primary metabolites in *Arabidopsis thaliana* under conditions of low and high photorespiratory activity. This was achieved by growing plants under elevated CO_2_ (1000 ppm) from the seedling stage and shifting them to ambient atmospheric CO_2_ (~450 ppm) for one diurnal cycle. Comparing flux calculations of both conditions revealed that a sudden increase in PR activity not only affected carbon assimilation, but also altered flux of carbon and nitrogen into the different families of amino acids. In the *hpr1-1* mutant of *Arabidopsis*, which is strongly restricted in the conversion of hydroxypyruvate to glycerate, this caused a marked shift in the flux from the oxoglutarate family to the aspartate family of amino acids that seems related to a modification in the leaf balance of asparagine (Asn) and Ser.

## 2. Results

### 2.1. Data Collection and Model Construction

The aim of this study was the identification of changes in flux of carbon and nitrogen into the different families of amino acids following changes in atmospheric CO_2_ concentration. To achieve this, 28 primary metabolites, i.e., starch, soluble sugars, carboxylic acids, amino acids, as well as ammonium, were quantified in intervals of 2 h during the light phase and once in the middle of the night in plants acclimated to eCO_2_ as well as in plants shifted from eCO_2_ to aCO_2_ for one full diurnal cycle. At five-fold replication, this yielded a data matrix of 1680 data points each for the Col-0 wildtype and the *hpr1-1* mutant. For each of the two growth conditions, a two-way ANOVA for the effects of genotype and daytime was performed with emphasis on the interaction term, which indicated differences in the diurnal dynamics between the genotypes. For the starting condition, eCO_2_, significant interactions were found for Ser, Gly and fumarate. Figure 1A–C shows the diurnal dynamics of these metabolites. It should be noted that the CO_2_ concentration of 1000 ppm reduces, but does not prevent, PR as already reported [8]. Thus, Ser accumulated in the *hpr1-1* mutant during the light phase, while its level showed slight variation in the wildtype (Figure 1A). Although at low absolute concentration, Gly accumulated in both genotypes during the day, but faster and to a higher level in *hpr1-1* (Figure 1B). As reported earlier, fumarate accumulated during the day in the wildtype, while it fluctuated at an elevated level in the mutant, showing no clear profile. When shifted to aCO_2_, Ser and Gly massively accumulated in the mutant during the day (Figure 1I,J), while glutamate (Glu) was consumed, probably as a source of ammonium for transamination of glyoxylate in the PR pathway. This was more intense in the mutant (Figure 1H). In conjunction with that, deviating profiles for soluble sugars were observed for wildtype and *hpr1-1*. While hexoses strongly accumulated in the mutant during the day (Figure 1L,M), sucrose accumulation was delayed and only half the amount of the wildtype (Figure 1N). For various other metabolites, genotype effects and/or significant differences of time points were observed, but there were no clear differences in the diurnal profiles. The results of the ANOVA are summarized in Appendix A.

For modeling, the data set was condensed to hold 12 groups of metabolites, which represented starch, soluble sugars (glucose, fructose, sucrose), two groups of carboxylic acids, i.e., malate plus fumarate (MF) and citrate (Cit), three groups of amino acids, i.e., the pyruvate family (Ala, Val, Leu) plus aromatic amino acids (Phe, Tyr, Trp) and Cys (AA1), the aspartate family (Ile, Thr, Asn, Asp, Met, Lys, termed AsF), a subset of the oxoglutarate family (Arg, Pro, termed KgF) and four individual amino acids Gly, Ser, Glu and Gln, as well as ammonium. The latter amino acids were observed individually because of their central role in PR and nitrogen assimilation, respectively. This resulted in the model shown in Figure 2. Net photosynthesis (NPS) was measured and used as an input into the system. At eCO_2_, mean NPS was 140.1 and 120.4 µmol g^−1^ h^−1^ for Col-0 and *hpr1-1,* respectively, during the day and −13.4 and −12.0 µmol g^−1^ h^−1^ during the night. At aCO_2_, NPS dropped to 115 and 110.1 µmol g^−1^ h^−1^ during the day and −14.2 and −12.5 µmol g^−1^ h^−1^ during the night. Based on compensation point data from [14], bounds for PR at eCO_2_ were set at 10 to 13% NPS and at 20–22% at aCO_2_. PR was subtracted from NPS to yield the carbon flux (1) into the sugar phosphate pool. At the same time, PR created the second input (7) that yields Gly as the first measured metabolite. Two molecules of Gly give one Ser and an output of one CO_2_. This is depicted as (8). Night respiration created the second output of CO_2_ (12). All other outputs from the system are exports of metabolites, either into compounds, which are beyond the scope of the model, e.g., protein, cell wall, lipids, secondary metabolites etc., or transported out of the leaf via the phloem. To account for the possibility that Ser could be imported into the leaf from the root system (see [11]), the lower bound of exp4 was set to negative values, which as such allows import into the leaf. This is shown by a double-sided arrow in Figure 2. For the conversion (8), two single-sided arrows were used, because the production of Gly from Ser is not a reversion of the Gly decarboxylation reaction: only one Gly is produced from one Ser, and the C1 is transferred to tetrahydrofolate (THF), but not released as CO_2_. Thus, two separate terms for this conversion were used in the ODE system. The same is true for starch build and degradation, because these reactions have different time frames: starch synthesis is set to zero during the night, while degradation is zero during the day. 

The model was translated into a system of ordinary differential equations (ODE) that describe carbon dynamics from uptake by photosynthesis to release by respiration or export of compounds from the plant shoot (Appendix A). To correctly capture stoichiometry, the C-content of metabolite groups was calculated based on the molecular formula and relative contribution of individual compounds to the pool. Nitrogen dynamics were linked to carbon dynamics in the same way by taking into account the N/C ratio of compounds and the mean N/C ratio of groups, respectively. All kinetics were modeled by mass action law. It should be emphasized at this point that, because only metabolite levels and a rate constant are considered, this approach does not allow investigating regulatory instances or enzyme properties. In addition, sub-cellular allocation is not taken into account, although it can severely influence effective concentrations [15,16]. Rather, simulations based on the model address gross fluxes of C and N and their diurnal dynamics depending on atmospheric CO_2_ concentration. The model ODE and constants are given in Appendix A. Model construction afforded inclusion of two additional states that, using the GC/MS-based method applied in this study, could not be quantified reliably. Hexose phosphates were inserted as the central hub for carbon supply, and oxoglutarate as an intermediate in the carbon supply to Glu synthesis. Values for hexose phosphates were taken from our previous work [17] as starting points for parameter identification. Oxoglutarate was a special case, because it is part of the only two-substrate reaction. Glu synthesis has been modeled from Gln as N- and a combined citrate/oxoglutarate pool as C-source in previous studies [14]. However, as in the present study, using mass action kinetics turned out problematic because of deviating concentration dynamics of the two substrates, which precluded computation of one reliable kinetic constant. This was bypassed by creating a “virtual” oxoglutarate state with dynamics adjusted to Gln that was built from citrate in a reaction with a separate kinetic constant (Figure 2). It should be noted that this is an artificial state, which was named “oxoglutarate” to ease comprehension of the pathway, but is not representative of in vivo oxoglutarate concentrations. 

### 2.2. Parameter Identification and Flux Calculations

Based on the ODE system given in Appendix A and graphically represented in Figure 2, vectors of kinetic rate constants were identified at intervals of two hours. These were supported by 12 metabolite levels determined at the same frequency during the day. Because the night consisted of only three measurements, the missing values were interpolated by a smoothing spline (see Section 2). The *paropt* package [18] for the R programming environment was used for parameter identification. Although *paropt* uses the particle swarm algorithm for parameter identification as a global optimizer, it was not possible to identify a suitable parameter set in one single step because of the complexity of the model. We thus applied the strategy of consecutively increasing model complexity that was introduced by [17]. Using four steps, we successively identified parameter upper and lower bounds for sugar and carboxylic acid metabolism, photorespiration, the different amino acid families and finally the GS/GOGAT cycle, before N fluxes were integrated. 

For the final model, suitable parameters were identified in at least 20 independent runs, reaching a value of the cost function at least smaller than 1. While metabolite dynamics were similar for Col-0 and the *hpr1-1* mutant in the case of starch and sugars, striking differences occurred not only for Ser, which accumulated to levels three-fold higher in the mutant during the day, but also for the MF pool and amino acids of the aspartate (Asf) and oxoglutarate (KgF) family (Figure 3). The former accumulated during daytime in Col-0, but not in the mutant. The latter stayed more or less constant in the mutants, but increased over the day in the wildtype. Similar dynamics were obtained for Gln.

When plants were shifted to ambient CO_2_, an interesting difference occurred for the allocation of carbon to starch and soluble sugars with lower levels of starch and higher levels of sugars in *hpr1-1* (Figure 4). The very steep increase in Gly and Ser in *hpr1-1* was accompanied by lower levels as compared to Col-0 of other amino acid families, especially the oxoglutarate family, and an increase in the MF pool. The combined pool of Cys, aromatic amino acids and the pyruvate family (AA1) showed opposite dynamics with an increase in the first half of the day in *hpr1-1*, but in the second half in Col-0. Another striking feature of the plants shifted to ambient conditions was that not only the dynamics of Ser and Gly had much higher amplitudes, but also those of Glu, Gln, the AA1 pool and the aspartate family, the latter especially in *hpr1-1*. Generally, simulations for sugars, the amino acid pools as well as Glu/Gln were more accurate than for carboxylic acids and also for Gly and Ser that showed deviations, especially at the day/night transition. 

With the identified parameters, flux could be assessed according to the mass action law, by simple multiplication of the actual kinetic rate constant and the metabolite concentration. It should be emphasized that, especially with the relatively rough grid of parameter estimation every 2 h, parameter values tended to show oscillation, because the cost function is computed only for the measured metabolite concentrations. Thus, calculated flux also oscillates. However, this is a systematic feature and thus comparison of genotypes and conditions is still possible. This revealed that, under elevated CO_2_, the flux into sugars was much higher in Col-0 than *hpr1-1*, while especially during the night the flux into Cit was higher in *hpr1-1* (Figure 5). Considering that even under the high CO_2_ condition of 1000 ppm, photorespiration still occurred, it is not surprising that the flux from Ser back to the sugar phosphate pool was higher in Col-0 during the day. However, the flux into Ser was the same for both genotypes. Looking at metabolite export, sugars, mainly sucrose, had the highest share in Col-0, but export of Ser during the day and aspartate family-derived amino acids during the night were higher in *hpr1-1*. Ser export attained negative values during the night, which means that Ser should have been “imported” either by the PPSB pathway or the vasculature. This was found for both genotypes at very similar levels.

After the shift to ambient CO_2_ concentration, the amplitude of sugar dynamics declined in Col-0 (Figure 6, f_sugar), while flux through the GS/GOGAT cycle was elevated. This was more pronounced in the wildtype as compared to the *hpr1-1* mutant, where the higher flux into the AsF pool persisted (Figure 6, f_asf). The same was true for the flux into the oxoglutarate family and to Ser and back to the sugar phosphates. Especially Ser, but also the AA1 pool, showed a pronounced day/night rhythm, as did the export of amino acids from the oxoglutarate family. Since Glu and Gln were separated, the oxoglutarate family here consists of only three amino acids, arginine, proline and histidine. Proline accounted for about 40% of this pool (see Appendix A). Remarkably, the flux from Ser to Gly, which depicts decarboxylation by serine-hydroxymethyl-transferase (SHMT) or transamination by Ser-aminotransferase [19], that was basically zero at elevated CO_2_, started operating after the sudden drop in the CO_2_ level. 

Effects of reduced CO_2_ were different in the *hpr1-1* mutant. Compared to the elevated CO_2_ condition, flux into sugars increased at the end of the day, as did flux into carboxylic acids. The flux into Ser was elevated. However, the flux back to sugar phosphates, which occurred to some extent at elevated CO_2_, appeared even further reduced at ambient CO_2_. Surprisingly, the calculated import of Ser during the night was much higher in *hpr1-1* at ambient CO_2_. At the same time, flux from Ser to Gly was strongly elevated. It is also interesting to note that export of amino acids from the oxoglutarate as well as the aspartate family was elevated in *hpr1-1* plants at ambient CO_2_. The most striking difference between the two genotypes was the higher turnover of Glu and Gln in the GS/GOGAT cycle in the wildtype and the observation that Col-0 exported more amino acids of the oxoglutarate family, while *hpr1-1* exported aspartate family amino acids at a higher level.

From the calculated flux values, a scheme of carbon allocation was drawn where the width of arrows indicates the integral of flux into a metabolite group either during the day (Figure 7A–D) or the night (Figure 7E–H). Elevated CO_2_ (Figure 7A,B,E,F) showed a higher proportion of flux into the aspartate family, while ambient CO_2_ caused a higher turnover of the GS/GOGAT cycle and a higher flux into the oxoglutarate family, especially in the wildtype. Figure 7 also shows that flux into sugars was higher in Col-0 (Figure 7A,E) as compared to *hpr1-1* (Figure 7B,F). 

## 3. Discussion

A sudden drop in atmospheric CO_2_ concentration is not likely to occur in nature, where CO_2_ levels constantly increase due to human consumption of fossil carbon stores. Nevertheless, the metabolic response of plants to such a condition allows for investigating interactions of primary metabolism and PR. PR is not only coupled to the production of amino acids Gly and Ser, but also affords turnover in the GS/GOGAT cycle for re-fixation of ammonium [20] and is thus directly connected with N-fixation [21]. While Ser is the main product leaving the PR cycle [22], the pathway is also directly connected to Cys and tryptophane synthesis, as well as to the production of secondary metabolites, e.g., glucosinolates. Less is known of interactions with sugar and carboxylic acid metabolism and how PR affects amino acid homeostasis. Here, we used a combined HPLC and GC/MS based metabolomics approach to quantify 28 primary metabolites that were followed over a complete diurnal to study their dynamics at either low or high photorespiratory activity. 

The shift from high to low CO_2_ caused the expected reduction in starch synthesis and a slower accumulation of soluble sugars, but led to a strong increase in the combined pool of amino acids from the pyruvate and aromatic amino acid families and Cys in the wildtype. Cys is the main constituent of this pool, and it is the Cys levels that cause this increase. A tight connection between increases in Ser and Cys was also observed by [23], who analyzed metabolites of the *hpr1-1* mutant grown at constant ambient CO_2_ concentrations. In the *hpr1-1* mutant shifted from high to low CO_2_, the accumulation of Cys was even faster. In the wildtype, the increase in Cys appeared at the cost of the oxoglutarate family, which showed accumulation over the entire day at elevated CO_2_, but only until 6 h into the light phase at ambient CO_2_. A very likely explanation is the higher demand of Glu in the GS/GOGAT cycle, but the flux calculations also point to a strong increase of export of oxoglutarate-derived amino acids, the main constituent being proline, concomitant with enhanced flux into this pool. This, in turn, was related to reduced flux into and export of aspartate family amino acids in the wildtype immediately after the shift to the lower CO_2_ level. However, this did not occur in the *hpr1-1* mutant that is limited in photorespiratory activity. It thus appears that increased PR causes a redirection of flux from the aspartate to the oxoglutarate family without accumulation of the latter in the shoot. Either protein turnover or export via the phloem could account for this lack of accumulation. It has been shown that substantial amounts of amino acids are exported to sinks during the day [24] and thus we speculate that phloem export of proline might be responsible for this flux. Enhanced amino acid production is supported by a larger Cit turnover in wildtype in ambient conditions, which was not observed for *hpr1-1*. Citrate is the main carbon source for amino acid synthesis and its decline during the light phase, which was also observed in our study (Figure 3 and Figure 4), indicates that it stems from stores built up during the previous night [25]. All the more astonishing was our finding of rather high flux into Cit during the light phase. However, [26] demonstrated for isolated pea leaf protoplasts that conditions of high PR activity, which cause overreduction of the mitochondrial NAD pool, stimulate citrate synthesis in mitochondria and export into the cytosol, where cytosolic isocitrate dehydrogenase supplies 2-oxoglutarate for the photorespiratory re-fixation of ammonia. This so-called “citrate valve” [27] appears to operate independent of the nocturnal build-up of a citrate pool in the vacuole [28]. Findings for *hpr1-1* seem to support this view: at ambient CO_2_ the Cit level was lower as compared to wildtype, while the flux was elevated, pointing to excessive consumption. The flux into Glu, which serves as N-donor for transaminating glycolate, was two-fold higher at ambient as compared to elevated CO_2_ in *hpr1-1.* This could result from a lack of return of N when the PR cycle is not closed. However, the flux into Glu increased more strongly in the wildtype under ambient conditions. One possible explanation is that most amino acid pools, except Ser and Gly, were higher in Col-0 than *hpr1-1*, which would afford higher Glu turnover, because Glu is the central hub for transamination reactions. In *hpr1-1*, instead, flux appeared to be focused on the aspartate family that was clearly dominated by asparagine (Asn). Asn is the major transport form of assimilated N from sources to sinks [29], and in accordance with this, export of the aspartate family attained levels twice as high as in wildtype. But why does *hpr1-1*, under ambient CO_2_, export Asn to the root?

To address this question, we suggest looking at Ser dynamics, which, under elevated CO_2_ conditions, still show an increase during the day in both genotypes. During the night, Ser declined in wildtype but remained at a higher level in *hpr1-1*. In both genotypes, Ser dynamics were coupled to an export during the day, when PR operates, but in the night this flux became negative, indicating uptake from the vasculature or production of Ser by the so-called “phosphorylated pathway” [10]. This pathway uses 3-phosphoglycerate from plastidial glycolysis as substrate and is the main Ser source in heterotrophic tissues [30]. Although PR is the main source of Ser in photosynthetic organs [31], it was demonstrated that plastidial 3-phosphoglycerate dehydrogenase (*PGDH1*), a key gene of the PPSB, is induced in leaves during the night. The authors of [32] demonstrated that a functional PPSB is essential for vegetative growth, and reduction in its activity causes slowdown of growth and accumulation of amino acids even in photorespiratory growth conditions. Recently, [33] demonstrated that the PPSB was even more important for providing Ser than PR. Although genes coding for PPSB enzymes appear constitutively expressed in shoot and root, the site of Ser production by the PPSB is not clear. Feeding experiments, where Ser was added to growth medium for the recovery of *pgdh1* mutant plants, have demonstrated that Ser provision to the roots could restore plant growth [33]. Complex organ interaction has been demonstrated for Cys and Ser metabolism in the genus *Flaveria*, where all stages of the transition from C3 to C4 photosynthesis are represented. The C4 metabolism that largely prevents PR affords a re-allocation of Ser and Cys synthesis from mesophyll cells to bundle sheath cells in monocotyledonous species [13]. However, in dicotyledonous C4 plants from the genus *Flaveria*, Cys synthesis takes place predominantly in the roots, where Ser is produced by the PPSB [11]. Although transcripts for ATP sulfurylase and APS reductase, involved in Cys provision for glucosinolate production, have been detected in *Arabidopsis* bundle sheath cells [34], it is possible that Ser homeostasis during the night involves provision by the root because of higher PPSB activity. If this was the case, transport of Asn to the root could equal the N-balance, thus explaining the high Asn export of *hpr1-1* plants under the elevated as well as ambient CO_2_ conditions. This hypothesis is supported by observations originally made by [32]. These authors found that the silencing of *PGDH1* in *Arabidopsis* resulted in accumulation of Asn in the shoot. The analysis was then refined by [33], who found that Asn accumulated also in the roots of *PGDH1*-silenced plants, especially at the end of the light period. At the same time, Ser levels were reduced in the root, pointing to the disruption of the amino acid exchange we hypothesize based on the presented flux calculations. Additional support comes from the observation that the backward flux from Ser to Gly, which was implemented in our model to represent the activity of serine-hydroxymethyltransferase (SHMT), was elevated in *hpr1-1* after the shift to ambient CO_2_. This flux was marginal at elevated CO_2_, and minimal levels were computed for the middle of the night (Figure 5). However, artificially restricting this flux in simulations for *hpr1-1* under ambient conditions caused Gly levels to immediately collapse after light off. Considering the high levels of Gly accumulated over the day, there seems no demand for additional production by SHMT. However, the calculated flux most probably represents a different reaction: Ser-glyoxylate aminotransferase (SGAT) catalyzes the conversion of glyoxylate and Ser into Gly and hydroxypyruvate [19], thus ensuring that glyoxylate would not accumulate. The SGAT gene was demonstrated to be induced at night and, in wildtype plants, stabilizes the Ser level [35]. In *hpr1-1*, however, it might counteract the accumulation of glyoxylate, stemming from glycolate that is produced by non-enzymatic oxidation of hydroxypyruvate in the peroxisome [36]. 

## 4. Materials and Methods

### 4.1. Plant Growth Conditions

*Arabidopsis thaliana* wildtype Col-0 and the *hpr1-1* mutant (SALK067724) were grown in soil (seedling substrate, Klasmann-Deilmann GmbH, Geeste, Germany) for 5 weeks in a growth chamber at 1000 ± 20 ppm CO_2_ with 8 h/16 h light/dark regime (100 µmol m^−2^ s^−1^; 22 °C) and fertilized every two weeks. Afterwards, half of the plants were harvested and the other half transferred to aCO_2_ (ca. 450 ppm) for one day and then harvested. At the harvest day, full rosettes of 5 plants per genotype were harvested every two hours during the light phase (time points 0, 2, 4, 6, 8) and once in the middle of the night (time point 16). Samples were frozen in liquid nitrogen, ground to a fine powder using a ball mill (MM200, Retsch, Haan, Germany) and stored at −80 °C until analysis. Net photosynthesis was measured using infrared absorbance as described [17]. PR activity was calculated based on measured compensation points as described by [14].

### 4.2. Metabolite Analysis

Sugars were measured by HPLC using a PA-1 column on a Dionex ICS600 system (Thermofisher, Dreieich, Germany) as described by [37]. Briefly, soluble sugars were extracted from ca. 25 mg of plant material into 80% ethanol at 80 °C. Extracts were dried and resuspended in 500 µL of distilled water. Residues of the ethanol extraction were solubilized in 1 mL 0.5 M sodium hydroxide at 95 °C for 30 min. After neutralization with 320 µL of 2 M acetic acid, the suspension was brought to 5 mM CaCl_2_, and 30 U of thermostable α-Amylase (Megazyme, Bray, Ireland) was added. Following 15 min at 95 °C, the suspension was acidified by adding 180 µL acetic acid, and starch was digested with 16 U of Amyloglucosidase (Megazyme) at 50 °C for 30 min. The glucose content was then determined via a coupled enzymatic assay resulting in oxidation of o-anisidine, which was measured photometrically at 540 nm. 

Carboxylic acids were extracted from ca. 25 mg of plant material into 1 mL of distilled water at 95 °C for 10 min. After centrifugation, samples were analyzed using a Dionex AS11-HC column (Thermofisher) as described by [17].

Amino acids were extracted into 1 mL of 10 mM HCl at room temperature (RT) for 10 min and bound to 10 mg DOWEX 50 WX2 resin (Sigma-Aldrich, Singapore). The resin was washed twice with 100 µL methanol to remove sugars, and amino acids were eluated with 150 µL 8 M ammonia/methanol (1/1). After vacuum drying, samples were derivatized with 50 µL N-tert.-butyltrimethylsilyl-N-methyltrifluoracetamide (MTBSTFA) containing 1% tert.-butyltrimethylchlorosilane (Sigma Aldrich) in 50 µL acetonitrile at 95 °C for 1h followed by 2 h incubation at RT. Amino acids were measured by gas-chromatography coupled to mass-spectrometry (GC-MS/MS). For injection, 1 µL of the derivatized sample was used. The GC-MS/MS device was a GCMS-TQ8040 (Shimadzu, Kyoto, Japan) using helium as carrier gas at a flow of 1.12 mL/min. The stationary phase was a 30 m Optima 5MS-0.25 µm fused silica capillary column. The injection temperature was 230 °C. The transfer line and ion source were set to 250 °C and 200 °C, respectively. The initial temperature of the column oven was 80 °C and this was increased by 15 °C/min until the final temperature of 330 °C was reached and held for 6 min. After a solvent delay of 4.6 min, spectra of the MS device were recorded in the Q3 scanning mode with specific target-ions for each metabolite. Ten nmol of Norvaline (Acros Organics) per sample were added as internal standard. Additionally, external standards were used for quantification.

Ammonium was quantified according to [38]. 

### 4.3. Data Evaluation and Modeling

Data evaluation was performed using the R software [39]. A two-way ANOVA for genotype and daytime effects were used for statistical analysis of the data presented in Figure 1. For parameter identification, missing values were calculated using the smooth.spline function of the R base package with setting spar = 0.3. Parameter optimization was performed in a way that in silico time courses best matched the measured time courses. For optimization and simulations, the R-package *paropt*, version 0.3.3, was used [18]. 

## 5. Conclusions

Applying a relatively cost-effective approach of recording diurnal metabolomics and mathematical modeling, we computed flux profiles for 12 metabolite groups that cover the most abundant primary metabolites. This revealed that changes in photorespiratory (PR) activity affect not only levels of carbohydrates and amino acids, but also the ratio of amino acid families which are not directly involved in the PR reactions. Specifically, members of the aspartate family accumulate when CO_2_ levels drop suddenly, while Cys levels show strong diurnal oscillations. Flux especially into Pro increases, when CO_2_ declines, but this is not reflected in an accumulation in the leaves. Export of amino acids from leaves was affected much more strongly than sugar export, demonstrating a profound impact of PR on the C/N balance of different plant organs. Considering that sink organs are widely used as food and feed, this points to substantial effects of climate change on crop quality, which should be analyzed further.

In the *hpr1-1* mutant that is restricted in the PR pathway, flux into Cit and the aspartate family dramatically increase when PR increases, and this may be due enhanced export of Asn in exchange of Ser that is imported into leaves during the night. Further research is needed to investigate possible exchange mechanisms for amino acids between different plant organs.

## Figures and Tables

**Figure 1 ijms-25-08394-f001:**
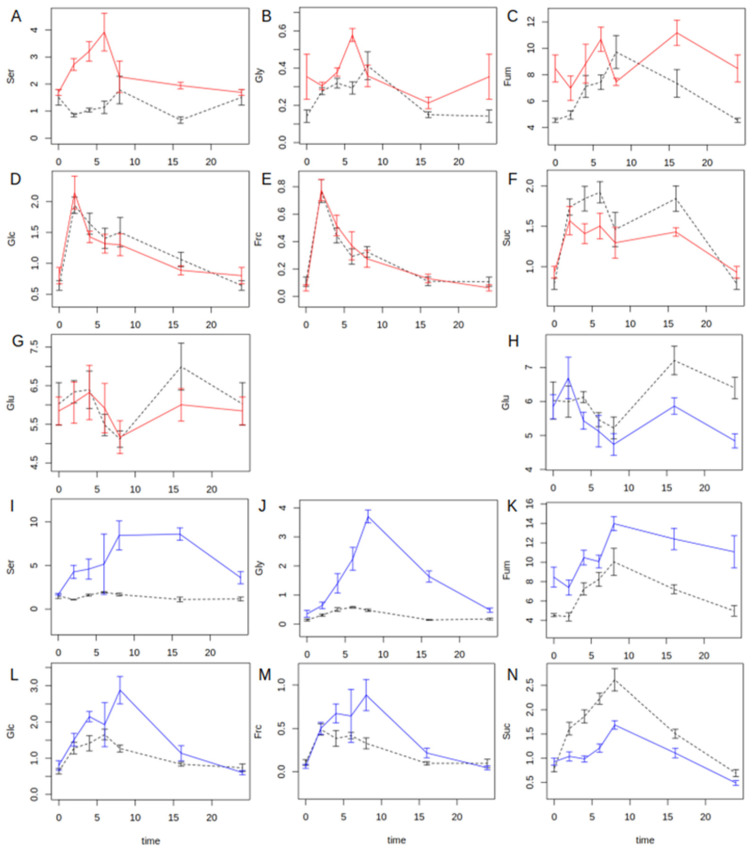
Diurnal profiles of metabolite concentrations in µmol g^−1^ fresh weight over time in hours from light on for metabolites that showed significant genotype × daytime interaction in a two-way ANOVA (*p* < 0.05, n = 5) at least under one CO_2_ treatment. Graphs show diurnal dynamics for Col-0 (black dashed lines) and the *hpr1-1* mutant (red lines for eCO_2_; blue lines for aCO_2_) at elevated CO_2_ (**A**–**G**) and after a sudden shift to ambient CO_2_ level (**H**–**N**). Lines connect means of measurements at time points 0, 2, 4, 6, 8, 16 and 24 h starting from light on. Light off was at 8 h. Error bars show standard error of the mean (n = 5). Results of the ANOVA for all metabolites are given in Appendix A.

**Figure 2 ijms-25-08394-f002:**
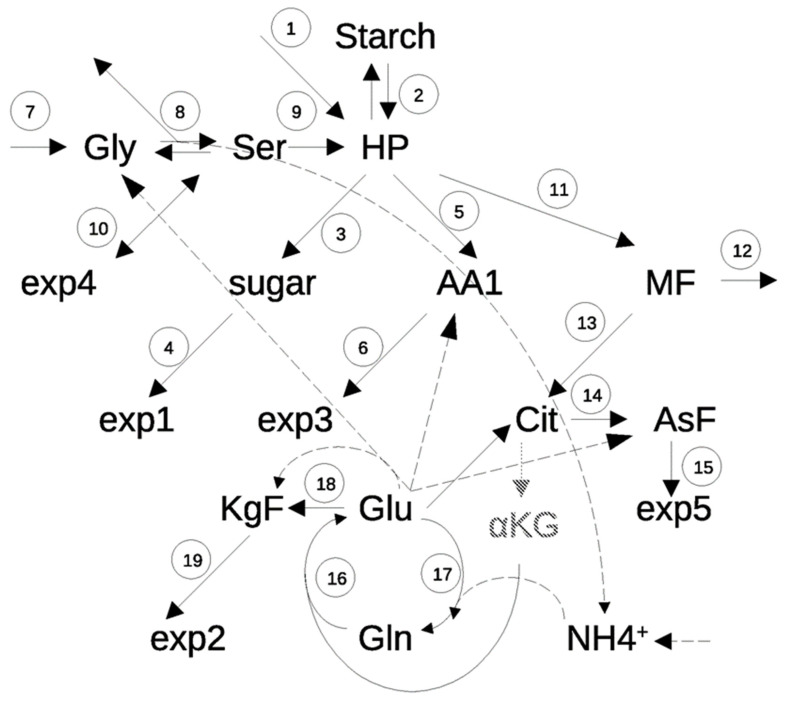
Model used for simulating primary metabolism. Black arrows show routes of carbon; dashed arrows show routes of nitrogen. Gly: glycine, Ser: serine; HP: sugar phosphates, ex4: export of Ser; AA1: pool of cysteine, pyruvate family and aromatic amino acids; MF: pool of malate and fumarate; exp1: sugar export, exp3: export of AA1, Cit: citrate, Asf: aspartate family; KgF: oxoglutarate family; Glu: glutamate; αKG: oxoglutarate, exp5: export of Asf; exp2: export of KgF, Gln: glutamine, NH4^+^: ammonium. Reactions 1: net photosynthesis; 2: starch build and degradation; 3: synthesis of sucrose, including hydrolysis to hexoses (sucrose cycling); 4: export; 5: synthesis of AA1; 6: export; 7: photorespiratory Gly production; 8. Gly decarboxylation and Ser synthesis; 9: hydroxypyruvate reduction; 10: export or import of Ser; 11: synthesis of malate and fumarate; 12: respiration; 13: synthesis of citrate; 14: synthesis of Asf; 15: export; 16, 17: GS/GOGAT cycle; 18: synthesis of KgF; 19: export.

**Figure 3 ijms-25-08394-f003:**
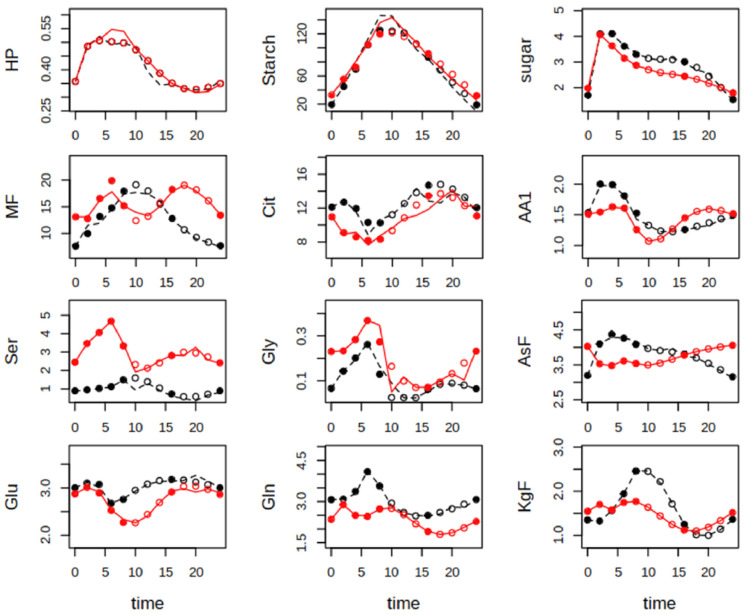
Measured and simulated metabolite levels in µmol g^−1^ fresh weight for Col-0 (black) and *hpr1-1* (red) plants grown constantly at elevated CO_2_ level (1000 ppm). Closed circles at time points 0, 2, 4, 6, 8, 16 and 24 are means of five-fold replication. Open circles were created by a smoothing spline (see Section 2). Lines are results of simulations with the best fitting parameter set. For abbreviations of metabolites, see Figure 2.

**Figure 4 ijms-25-08394-f004:**
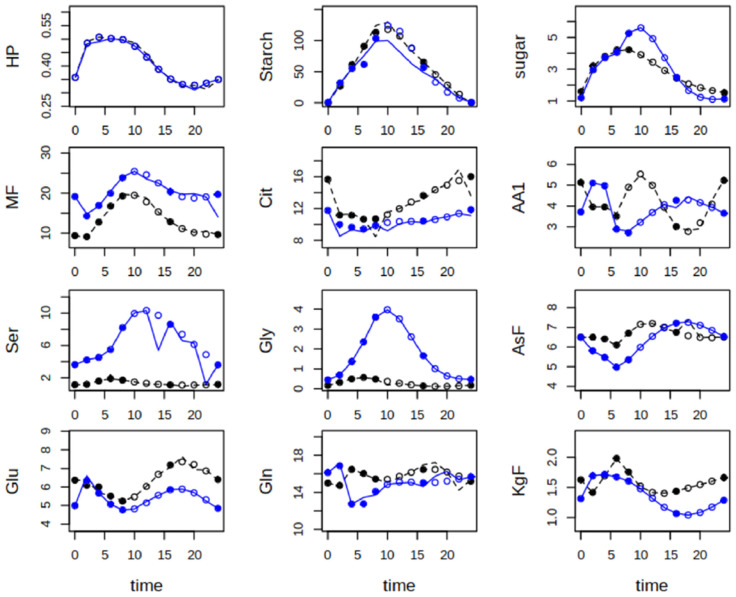
Measured and simulated metabolite levels in µmol g^−1^ fresh weight for Col-0 (black) and *hpr1-1* (blue) plants shifted from elevated to ambient CO_2_ level (~450 ppm). Closed circles at time points 0, 2, 4, 6, 8, 16 and 24 are means of five-fold replication. Open circles were created by a smoothing spline (see Section 2). Lines are results of simulations with the best fitting parameter set. For abbreviations of metabolites, see Figure 2.

**Figure 5 ijms-25-08394-f005:**
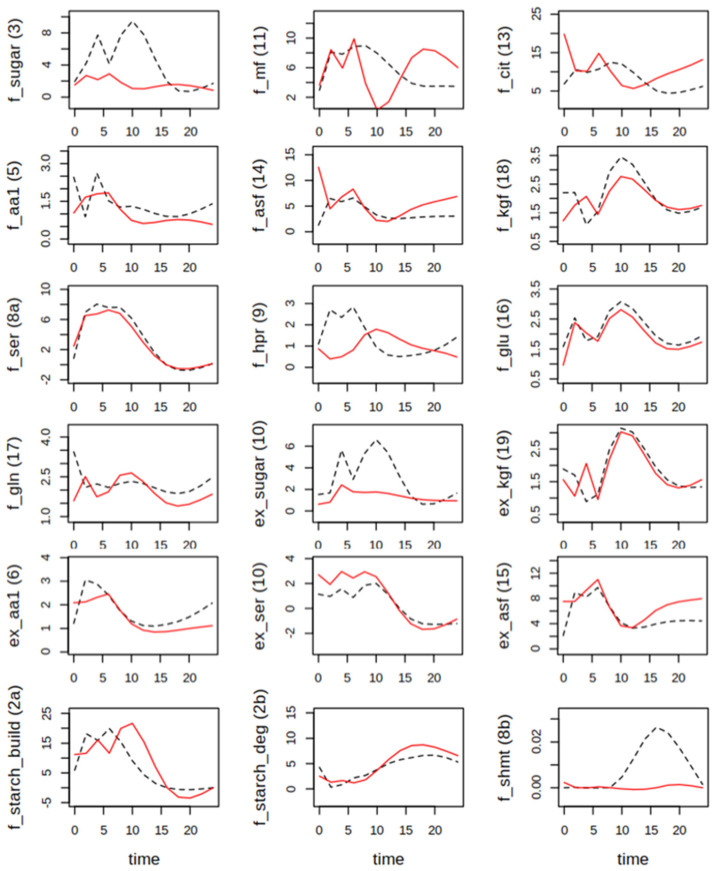
Calculated flux trajectories in µmol g^−1^ h^−1^ for Col-0 (dashed black lines) and *hpr1-1* (red lines) constantly grown at elevated CO_2_ level (1000 ppm). All fluxes are termed based on reaction products, except f_hpr, which is the HPR reaction, and f_shmt, which is a proxy for reactions leading from Ser to Gly. All exports out of the model scope have the prefix ex_. Numbers in parentheses refer to the numbering of reactions in Figure 2.

**Figure 6 ijms-25-08394-f006:**
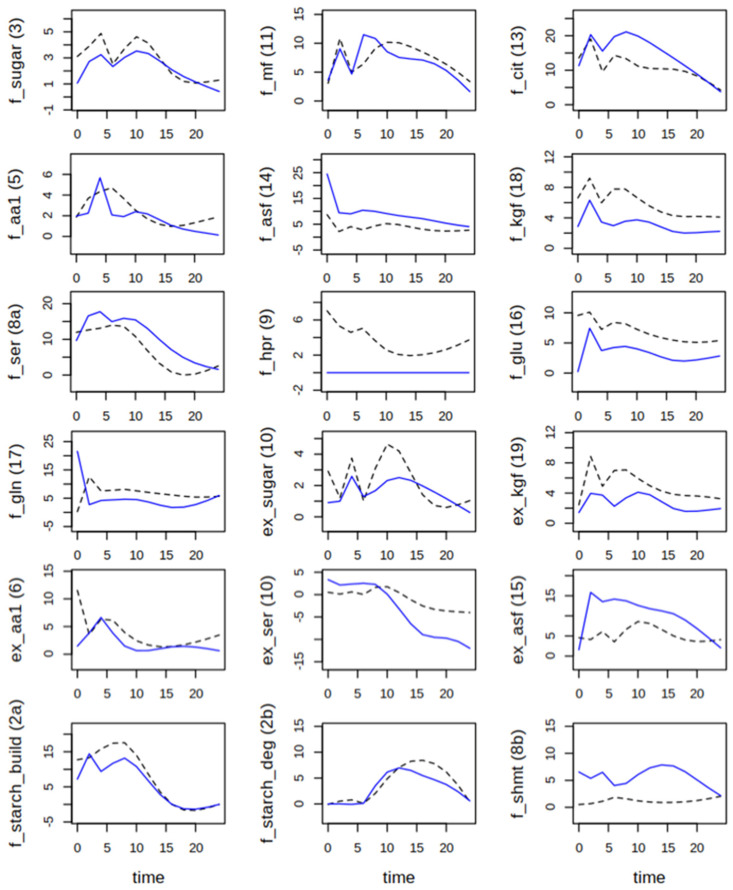
Calculated flux trajectories in µmol g-1 h-1 for Col-0 (dashed black lines) and hpr1-1 (blue lines) shifted from elevated to ambient CO_2_ level (~450 ppm). All fluxes are termed based on reaction products, except f_hpr, which is the HPR reaction, and f_shmt, which is a proxy for reactions leading from Ser to Gly. All exports out of the model scope have the prefix ex_. Numbers in parentheses refer to the numbering of reactions in Figure 2.

**Figure 7 ijms-25-08394-f007:**
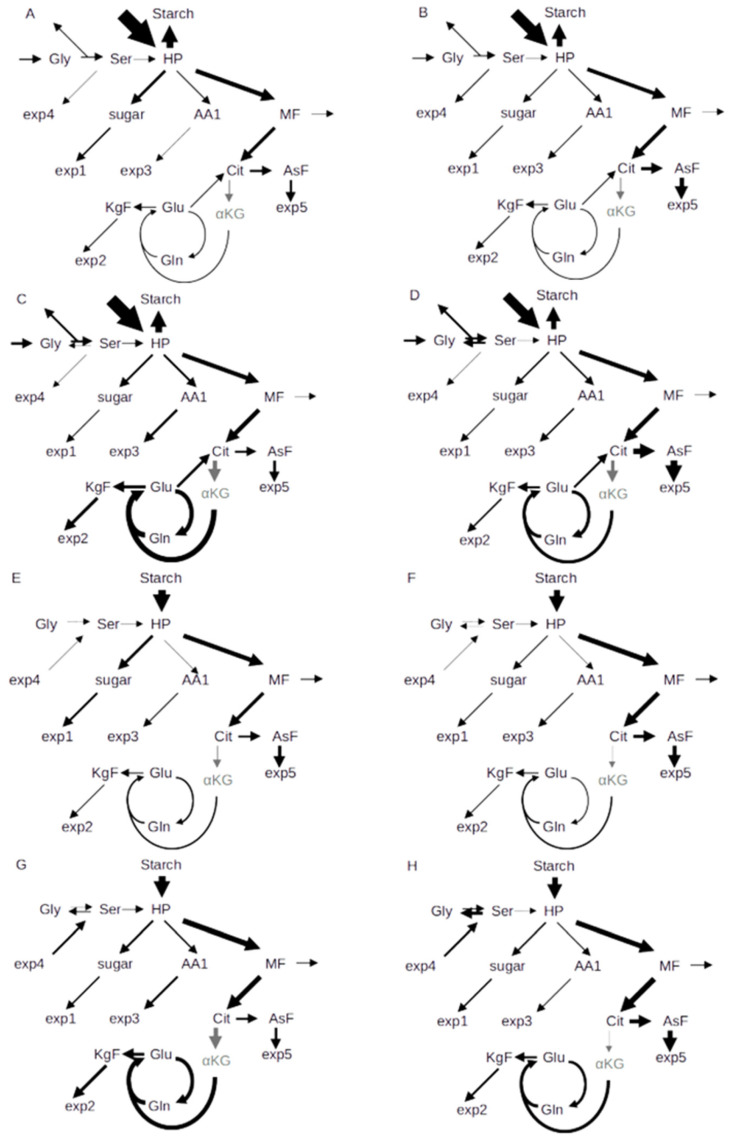
Schemes of carbon allocation during day (**A**–**D**) and night (**E**–**H**) for Col-0 (right) and hpr1-1 (left). (**A**,**B**,**E**,**F**): eCO_2_; (**C**,**D**,**G**,**H**): aCO_2_. The width of arrows indicates flux into a metabolite pool integrated over day and night, respectively.

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
