# Peer review of "Flux Calculation for Primary Metabolism Reveals Changes in Allocation of Nitrogen to Different Amino Acid Families When Photorespiratory Activity Changes"

_ijms, 2024, doi:10.3390/ijms25158394_

Round 1

Reviewer 1 Report

Comments and Suggestions for Authors

I find the study of interest and sound. Hence, I have only some minor comments to the authors attention.

Figure 1 should contain the same metabolite patterns for both CO2 conditions to improve comparability. Worth using colors (all figures) to differentiate the wildtype from the mutant better, i.e. increase visibility.

Conclusion section could be optimized in order to provide a more general interpretation of the overall dataset instead of repeating a few very specific interpretations. How does this study enhance our current understanding on the role of PR for overall leaf-metabolism? A slight outlook could be implemented to kick off new experiments based on the presented finding.  

There are certain typos in the current manuscript draft that could be picked up by careful re-editing such as:

Line 362: …, under elevated condition,…  - add CO2

Line 366: … vaculature … - vasculature?

Line 375: … genes encoding for PPSB enzymes

Line 403: … Ser-Glyoxylate Aminotransferase (SGAT)

Line 405: asuring – ensuring ?

Comments on the Quality of English Language

Some typos need to be corrected.

Reviewer 2 Report

Comments and Suggestions for Authors

This manuscript describes measurements of metabolites of samples taken every two hours either with or without a switch from growth CO2 of 1000 ppm to 450 ppm. Wildtype plants and the hpr1-1 mutant was used. Actual fluxes were not measured, for example it should be possible to measure photosynthetic rate and estimate photorespiration rate in the two conditions. I also think data should be shown telling how much of the decline in photosynthetic rate was the result of increased photorespiration rate and how much was independent of the change in photorespiration rate.

Figure 2 is the heart of the report but it is not very intuitive. There appear to be two inputs, CO2 (1) and glycine (7). It is hard to see glycine as a significant input to the system. If that glycine is from photorespiration why is there no path from the carbon input to the glycine input? It is not clear what is being proposed for serine export (exp4 and the arrow at 8). What is the difference between a double headed arrow (10) and two arrows in opposite directions  (2). Is starch synthesis considered reversible or in one direction during the day and the other direction at night?

Line 336 I would consider a full day to be a rather long time frame for much of this metabolism. What is the turnover time for the amino acid pools? How quickly are amino acids either made into protein or exported?

I don’t believe the report advances our understanding of metabolism very much.

Minor things

All the 2’s of CO2 need to be subscripted. It is harder to read when this is not done.

Line 48 – in many cases the reason for lower photorespiration in C4 is the increased CO2 more than reduced oxygen.

55 Please write out the words for PPSB.

56 do not capitalize names of molecules 3-phosphoglycerate, not 3-Phosphoglycerate

56 phosphoglycerate not phosphyglycerate

57 proportion (o missing)

63 and not und

65 homeostasis (spelling)

66 the phrase “made use of mathematically modelling” is not needed

94 spacing - 2 M not 2M

94 subscript the 2 of CaCl2

95 do not capitalize amylase

100, 142, 295 carboxylic acids (carbonic acid is H2CO3)

106  8 M (spacing)

Figure 1 do not use* for multiplication except in computer programs.

Figure 1 gram should be raised to the -1, not weight.

Figure 1 what are the units of time

Figure 1 can the labels be made bigger so they are easier to read?

146 I don’t see the need for the word “respectively”

150 the figure shows the data, the profiles, I would not call it a scheme of profiles.

179 molecular (spelling)

199 built not build

Figure 3 the measured versus calculated points should be distinguished on the graph, not just the legend.

Figure 2 is not very intuitive. There appear to be two inputs, CO2 (1) and glycine (7). It is hard to see glycine as a significant input to the system. If that glycine is from photorespiration why is there no path from the carbon input to the glycine input? It is not clear what is being proposed for serine export exp4 and the arrow at 8.

312, 313 can this sentence be revised to avoid using also twice?

313 e.g. pretty much means lie so like e.g. is repetitive

314 and 385 glucosinolates (spelling)

318 the reduction in starch is likely more the result of reduced photosynthetic rate even if this is caused by increased photorespiration. In other words, I would have expected the drop in starch synthesis even if the experiment were done in low oxygen removing the photorespiration issue

336 vasculature (spelling)

371 leaves not leaved

374-5 although (spelling)

376 “At least” not needed

386 double negative (not unlikely) should be avoided

404 I think that the data in South et al in Science, in which glyoxylate levels were very high in the plants that yielded better, shows that it is 2-PG, not glyoxylate, that is toxic.

442 and throughout the references there are many issues. If the journal has a copy editor or if the references will be set from DOIs this is not a problem, otherwise these all need to be fixed.

Comments on the Quality of English Language

This manuscript is poorly prepared making it unpleasant to read. The model they present in Figure 2 is confusing, not illuminating. I do not think it can be fixed.

Reviewer 3 Report

Comments and Suggestions for Authors

This study analyzed the diurnal changes of major metabolites under high and ambient CO2 concentrations in both wild type and a photorespiratory hpr1-1 mutant. While the topic is interesting, modeling or calculating the metabolic flux remains challenging without stable isotope labeling strategy. Data were collected from two CO2 concentrations that either inhibit or maintain photorespiration in the wild type and hpr1-1 mutant. However, multiple factors such as diurnal patterns, CO2 treatments, and genotypes significantly increase the complexity of the work. This high complexity prevents readers from fully understanding the authors' intentions. I have a few questions regarding the data presentation and interpretation:

1.     The authors claim the two-way ANOVA tested the genotype and treatment interaction. However, the p-values of ANOVA from Tables S1 and S2 were determined for genotype and daytime. How can daytime represent treatment with different CO2 concentrations? How is the genotype and treatment interaction derived from this analysis? The authors need to compare high and low CO2 concentrations for the genotype*treatment interaction.

2.     It is difficult to relate the model in Figure 2 to Figures 3-6, which challenges the understanding of the entire manuscript. The authors need a better way to show the relationship between the model and the actual/simulated data.

3.     There are three major conclusions in the abstract. The authors claim proline export from leaves increases, but no actual data is shown for proline; only speculation is mentioned in the manuscript (lines 338-339). The claim of an immense increase in turnover in the cyclic reaction of GS/GOGAT is unclear; it is obviously not the turnover of the GS/GOGAT enzyme. The substantial alteration in flux in the hpr1-1 mutant is also not clearly presented.

4.     Overall, the presentation of actual data and simulated flux needs substantial improvement. Using arrows with the intensity of flux could better present the changes in two separate models (CO2 concentration and genotype). Separating CO2 concentration treatment and genotype with metabolic flux changes could improve the study's readability for non-expert readers outside the photorespiratory field.

Minor issues:

How could the decline in CO2 happen 35 billion years ago (line 46)? The Earth has only existed for about 4.5 billion years.

Comments on the Quality of English Language

The authors should write clearly what is speculation in the abstract and other parts.

Round 2

Reviewer 2 Report

Comments and Suggestions for Authors

The authors have responded well to my comments. Two issues that should be addressed

1. RuB is an unusual abbreviation for RuBP. I suggest using the well established abbreviation. (line 34)

2. Figure 3 and 6 are not cited in order in the text.

Reviewer 3 Report

Comments and Suggestions for Authors

The revised manuscript measured the net photosynthesis and supply a model to help readers follow the major flux changes in both treatment-control and/or wild type-mutant comparisons. However, I still have a few concerns.

1.     Arabidopsis plants involving wild type and hpr1-1 mutant should be shown to illustrate the experiment setup.

2.     Protein turnover studies suggest Arabidopsis have a degradation rate around 10% per day. Protein synthesis would be much higher for growing plants. So it is not impossible that the turnover changes involving both degradation and synthesis would contribute to the amino acid pool. The comments from line337-342 should consider the protein turnover in a different way.

3.     The authors should make it clear in the abstract that export of proline is speculated rather than a fact i.e. Line 290-292,line 331-334.

4.     Considering the nature of this study, it would be helpful to refine the major findings rather than presenting too many speculations.

Comments on the Quality of English Language

It could help the readers to follow the major findings with some extra explanation of the assumptions in the manuscript.  
